# Radiation Promotes Acute and Chronic Damage to Adipose Tissue

**DOI:** 10.3390/ijms26125626

**Published:** 2025-06-12

**Authors:** Kia T. Liermann-Wooldrik, Elizabeth A. Kosmacek, Joshua A. McDowell, Simran Takkar, Divya Murthy, Pankaj K. Singh, Micah B. Schott, Moorthy P. Ponnusamy, Rebecca E. Oberley-Deegan

**Affiliations:** 1Department of Biochemistry and Molecular Biology, University of Nebraska Medical Center, Omaha, NE 68198, USA; 2Department of Pathology and Laboratory Medicine, Brown University, Providence, RI 02912, USA; 3Eppley Institute for Research in Cancer and Allied Diseases, University of Nebraska Medical Center, Omaha, NE 68198, USA; 4Department of Oncology Science, The University of Oklahoma Health Sciences, Oklahoma City, OK 73104, USA

**Keywords:** adipose tissue, radiation, oxidative stress, metabolism, inflammation, senescence

## Abstract

Radiotherapy is commonly used for treating various types of cancer. In addition, adipose tissue is not routinely spared during typical radiation treatment. Although radiation is known to induce metabolic effects in patients, the effects of radiation therapy on adipose tissue have not been elucidated. Currently, few studies have investigated the impact of radiation exposure on adipose tissue, and these have primarily involved whole-body irradiation. This study aimed to understand the acutely persistent damage caused by clinically relevant radiation doses in adipocytes. Specifically, in vitro and in vivo, irradiated adipocytes increased reactive oxygen species (ROS) and lipid peroxidation levels and elevated lipolytic activity compared to unirradiated adipocytes. RNA sequencing also revealed the upregulation of senescence and inflammation pathways. We observed an increase in macrophage and T-cell accumulation at both 1 and 6 months after radiation exposure using in vivo models. Many of the changes observed in irradiated adipose tissue, including oxidative stress, metabolic dysfunction, inflammation, and senescence, are consistent with those observed in adipose tissue from obese patients, in which obesity is a known driver of many cancers. As adipose tissue damage is maintained chronically, protecting adipose tissue from the harmful effects of radiation exposure may improve radiation-induced toxicity and reduce cancer recurrence and progression.

## 1. Introduction

Adipose tissues, both visceral and subcutaneous, are among the most abundant tissues in the human body. Adipose tissue is highly complex and composed not only of adipocytes but also of preadipocytes, immune cells, extracellular matrix, and stromovascular cells, with this complexity notably conserved across species [1]. Adipose tissue is metabolically active and functions to store excess lipids, secrete adipokines, and regulate thermogenesis. Owing to the secretory nature of adipose tissue, which releases adipokines, hormones, and lipids, it has been widely accepted as an endocrine organ [2]. Adipose tissue health has been implicated in a variety of diseases, including diabetes [3], cardiovascular health [4], hepatic steatosis [5], and some cancers [6].

Radiation therapy is a commonly used treatment option for several cancer types and stages, including brain [7], head and neck [8], colorectal [9], cervical [10], prostate [11,12], and breast cancer [13]. Radiation damages or slows the growth of cancer cells by directly damaging DNA through double-strand breaks or indirectly causing DNA damage through the generation of reactive oxygen species (ROS) [14]. When patients are treated with radiation, it is nearly impossible to avoid hitting neighboring non-cancerous tissues. Exposure of healthy tissues to radiation causes both acute and chronic damage, leading to unwanted side effects, including fibrosis. Our lab has previously shown that healthy, unirradiated adipose tissue protects against radiation-induced toxicity through adiponectin secretion [15,16]. However, adipose tissue is generally unavoidably affected during radiation treatment.

Adipose tissue is found near the sites of many tumors that are treated with radiation therapy, such as abdominal, pelvic, and breast cancers. Current radiation planning does not account for or spare this adipose tissue. Clinical data suggest that radiation to adipose-rich regions is detrimental, leading to an increased risk of cancer recurrence and metastasis compared to non-radiation interventions [17,18,19]. Few studies have investigated the impact of radiation on adipose tissue [20]. Most notably, Poglio et al. documented that in mice, a single dose of total body irradiation (7 Gy or 10 Gy) induced acute damage to subcutaneous adipose tissue by specifically damaging adipose lineage cells [21]. However, the effects of radiation using clinically relevant radiation dosing have not been determined, nor has the impact of radiation on visceral adipose tissues been described.

Since adipose dysfunction is linked to many different diseases and disorders, this study aims to describe the alterations in adipose tissue following a hypofractionated dose of radiation, such as that observed in the clinic. In this study, oxidative stress, metabolism, inflammation, and senescence were investigated in vitro and in vivo 1, 2, and 6 months post-radiation exposure, revealing that radiation causes chronic damage to adipose tissue.

## 2. Results

### 2.1. Adipocytes Are Oxidatively Stressed Following Radiation Exposure

Radiation is known to cause an accumulation of intracellular ROS, leading to the dysregulation of cellular processes. Utilizing 3T3-L1 adipocytes, we assessed the levels of ROS in adipocytes 48 h after exposure to 3 Gy of radiation for 3 days (RAD) or 0 Gy for 3 days (CON). Using immunofluorescence, we observed that irradiated adipocytes showed a significant increase in positive staining for 8-hydroxydeoxyguanosine (8-OHdG), indicating DNA/RNA oxidation [Figure 1A]. Since adipocytes store excess lipids in their lipid droplets, we suspected that lipid peroxidation levels would also increase as a result of radiation. Significantly increased lipid peroxidation was observed using an antibody against lipid peroxide 4-hydroxynoneonal (4-HNE) in irradiated adipocytes [Figure 1B]. Evidence of oxidative stress was further validated by flow cytometry. Total ROS and superoxide [Figure 1C] were quantified using a dihydroethidium (DHE) probe, and MitoSOX was used to quantify mitochondrial superoxide [Figure 1D]. Lipid peroxidation was measured using a BODIPY 581/591 C_11_ probe [Figure 1E]. Total ROS, superoxide, mitochondrial superoxide, and lipid peroxidation were all significantly enhanced in adipocytes 48 h after treatment with 3 Gy of radiation for 3 days. Excessive oxidative stress can compromise cellular viability, particularly when the dose of radiation causes irreparable DNA damage [22,23]; therefore, the viability of adipocytes was assessed 48 h after radiation. Interestingly, adipocyte viability, as determined by trypan blue exclusion, was not affected by radiation exposure [Figure 1F]. These results indicate that irradiation of adipocytes with 3 Gy radiation for 3 days induces oxidative stress but does not cause adipocyte cell death.

### 2.2. In Vitro, Irradiated Adipocytes Undergo Metabolic Dysfunction

Adipocytes maintain a large pool of neutral lipids within the lipid droplet [24]; these lipid stores were visualized using Oil Red O, a lipophilic dye. Oil Red O staining demonstrated that lipid droplet size significantly reduced with radiation treatment (3 Gy × 3 days) compared to unirradiated adipocytes (0 Gy × 3 days), suggesting that radiation causes a decrease in lipid stores [Figure 2A]. Similarly, in adipocytes stained with BODIPY and subsequently treated with or without radiation, we observed a decrease in lipid accumulation 48 h after irradiation [Figure 2B]. The decreased lipid accumulation observed after radiation indicated that lipolysis was potentially increased following radiation exposure. To further corroborate this, we stained adipocytes with BODIPY and irradiated them. Immediately following the last dose of radiation, the medium was replaced with fresh medium. Twenty-four hours later, fatty acid release was quantified by measuring the fluorescence intensity within the conditioned media of CON and RAD adipocytes [Figure 2C]. Forskolin, a known lipolysis inducer, served as a positive control [25]. RAD adipocytes had more BODIPY fluorescence within the media, indicating that following radiation, adipocytes undergo increased lipolysis. End products of lipolysis include free fatty acids. Seventy-two hours after radiation, free fatty acid levels were quantified from the conditioned media of the CON and RAD adipocytes. The free fatty acids were elevated after radiation, confirming that radiation induces lipolysis in adipocytes [Figure 2D].

Elevated levels of oxidative stress, such as those observed following radiation, cause alterations in gene expression and pathway activation [26]. Hormone-sensitive lipase (HSL) is a lipolytic enzyme that is phosphorylated by protein kinase A, leading to the activation of HSL [27]. To determine whether radiation affects the activation of lipolytic enzymes, the protein expression of phosphorylated (Ser565) versus total HSL was quantified via western blotting, with Ponceau serving as the loading control [Figure 2E]. Radiation causes an increase in phosphorylated HSL to total HSL in adipocytes. A fluorometric lipase activity kit was used to validate the increase in lipolytic activity of adipocytes 24 h after radiation treatment [Figure 2F]. Lipid droplet stores are maintained through a balance between lipolytic and lipogenic pathways. Fatty acid synthase (FASN) is the predominant enzyme involved in de novo lipogenesis and exhibited no significant alteration in 3T3-L1 adipocytes 24 or 48 h after radiation treatment; however, a trend of increased expression was observed at 48 h [Figure 2G]. These data show that following irradiation, adipocytes have heightened lipolytic activity but no significant increase in FASN expression, causing a reduction in lipid stores and overall lipid droplet size in vitro.

### 2.3. Adipocytes Undergo Senescence After Radiation Exposure In Vitro

To detect any transcriptional changes that adipocytes undergo when exposed to radiation, RNA sequencing data were acquired from adipocytes isolated from the gonadal adipose tissues of mice two months after receiving radiation (7.5 Gy) for five days to the pelvis [Appendix A. Several of the upregulated transcripts regulate inflammation, including *Il11*, *Cxcl10*, and *Tnfsf15*. Interestingly, *Cdkn1a*, otherwise known as p21, a cyclin-dependent kinase inhibitor and well-accepted marker of cellular senescence, was significantly elevated after radiation [28].

Cellular senescence is believed to help maintain tissue integrity upon DNA damage, such as after radiation exposure [29,30,31]. The tumor suppressor gene p16, an inhibitor of cyclin-dependent kinase (CDK) 4/6, is upregulated in senescent cells [32]. In the 3T3-L1 adipocytes, p16 gene expression was upregulated seven days after exposure to 3 Gy of radiation for three days [Figure 3A]. In addition to p16, the gene expression of p21 in irradiated adipocytes was significantly elevated compared to that in unirradiated adipocytes [Figure 3B], which confirmed the RNA sequencing data. To investigate protein expression, western blots were performed for p21, which exhibited increased expression in adipocytes exposed to radiation compared to unirradiated adipocytes [Figure 3C]. Upon senescence, cells produce higher levels of β-galactosidase (β-gal) [33]. Seven days post-radiation, the irradiated adipocytes demonstrated increased β-gal staining compared with the unirradiated adipocytes [Figure 3D]. Taken together, the elevated expression of p16 and p21 and the presence of β-Gal^+^ cells suggest that in vitro radiation exposure induces cellular senescence in adipocytes.

### 2.4. Adipose Tissue Sustains Oxidative Damage In Vivo

To characterize the impact of radiation exposure on adipose tissue in vivo, the gonadal fat pads of C57Bl/6 mice were exposed to 7.5 Gy of radiation for 5 consecutive days using the SARRP irradiator. Gonadal fat pads were chosen because they closely resemble human visceral adipose tissue [34]. The mice were sacrificed one month after radiation exposure, and gonadal fat pads were harvested. Adipose tissue sections collected from unirradiated (CON) and irradiated (RAD) mice were stained with antibodies against 8-OHdG and 4-HNE to quantify oxidative stress. As observed in vitro, DNA/RNA oxidation was significantly increased in adipose tissue collected from irradiated mice compared to unirradiated controls [Figure 4A]. Lipid peroxidation also significantly increased in the irradiated adipose tissue compared to the control [Figure 4B].

### 2.5. Adipose Tissue Maintains Oxidative Damage 2 Months Post-Radiation

Similarly, mice receiving pelvic radiation (7.5 Gy) maintained oxidative stress 2 months after radiation treatment, as observed by significantly increased DNA/RNA oxidation [Figure 5A] and lipid peroxidation [Figure 5B] in irradiated mice compared to unirradiated controls. H&E staining of the CON and RAD adipose tissues 2 months post-radiation demonstrated a decrease in adipocyte size in the irradiated adipose tissues [Figure 5C]. Since the lipid droplet occupies the majority of the adipocyte, this reduction in adipocyte size corroborated our previous finding that radiation reduces adipocyte lipid droplet size in vitro.

### 2.6. Immune Infiltration Is Increased in Irradiated Adipose Tissue

Adipose tissue contains resident immune cells, including macrophages and T-cells [35]. These immune cells are typically anti-inflammatory and help maintain healthy adipose tissue function [36]. RNA sequencing data suggested that irradiated adipose tissue might be inflamed [Appendix A. CD4+ T-cells secrete cytokines to signal immune infiltration, whereas CD8+ T-cells are cytotoxic and promote the recruitment of macrophages to adipose tissue [37]. Immunofluorescence staining of immune cells was performed on gonadal adipose tissue sections of male C57Bl/6 mice. One month post-exposure to 7.5 Gy for 5 days, a notable elevation in CD4+ helper T-cells [Figure 6A] and CD8+ cytotoxic T-cells was detected in irradiated adipose tissues [Figure 6B]. To maintain healthy tissue homeostasis, adipose tissue contains resident macrophages. To quantify both resident and non-resident macrophages, an antibody against F4/80 was used. Overall, macrophages in adipose tissue significantly increase and form crown-like structures one month after radiation exposure [Figure 6C]. Overabundant immune infiltration and crown-like structure formation in adipose tissue are hallmarks of unhealthy adipose tissue, as observed in individuals with insulin insensitivity or obesity [38,39,40,41]. Following radiation exposure, pro-inflammatory immune cell infiltration remains present at 1 month post-radiation.

### 2.7. Chronic Oxidative Damage Is Observed in Irradiated Adipose Tissue

Adipose tissue can sustain chronic damage, such as that observed in obesity, where oxidative stress, immune infiltration, and metabolic dysregulation are long-term problems. Additionally, burn victims can experience prolonged effects on white adipose tissue, including elevated metabolism years later [42]. Therefore, we hypothesized that the damage observed one and two months after radiation exposure would be chronically persistent. To observe chronic adipose damage, C57Bl/6 mice were irradiated with 7.5 Gy of radiation to the pelvis for 5 days. Six months after radiation, the gonadal fat pads were harvested. Interestingly, even after 6 months, irradiated (RAD) adipose tissue maintained elevated levels of oxidative damage, 8-OHdG [Figure 7A], and exhibited significantly higher levels of lipid peroxidation, 4-HNE [Figure 7B], than unirradiated (CON) adipose tissue. Therefore, adipose tissue is highly susceptible to and remains in a state of chronic oxidative stress after exposure to clinically relevant fractionated radiation.

### 2.8. Irradiated Adipose Tissue Is in a State of Chronic Metabolic Dysfunction

By weighing the adipose tissue of the mice and normalizing it to the total body weight of the mice, we compared the ratio of fat to total body mass in the control mice and mice that received pelvic radiation. Six months after radiation exposure, mice that received radiation had a significantly lower fat-to-body mass ratio than the unirradiated control mice [Figure 8A]. Adipose tissue maintains systemic metabolism by regulating lipid homeostasis. When adipose tissue undergoes lipolysis, triglycerides stored within the lipid droplet are hydrolyzed into free fatty acids and free glycerol and then released by adipocytes [43]. In mice that received pelvic radiation, serum free glycerol was significantly elevated [Figure 8B] and free fatty acids [Figure 8C] tended to be enhanced compared with the serum of unirradiated mice. Elevated serum fatty acid levels are one of the five conditions used to classify metabolic syndrome. Along with the reduced fat to total body weight ratio, these data indicate that radiation induces lipase activity and mobilizes free glycerols and fatty acids up to 6 months post-radiation.

Following radiation treatment, persistent metabolic impairment has been noted in cancer patients [44,45]. One hypothesis for radiation-induced metabolic diseases is the role of radiation in inducing mitochondrial dysfunction [45,46]. To investigate the overall metabolic health of irradiated adipose tissue, mature adipocytes were isolated from gonadal adipose tissues harvested six months after post-radiation exposure. Mature adipocytes were cultured for three days before metabolic capacity was assessed using Seahorse analysis. To test the responsiveness of the mitochondria, forskolin was used to stimulate lipolysis. The oxygen consumption rate (OCR) was depleted by inhibiting ATP synthase with oligomycin before being stimulated by carbonyl cyanide 4-(trifluoromethoxy)phenylhydrazone (FCCP) to induce maximal respiration. Rotenone/Antimycin then inhibited complex I and complex III to shut down mitochondrial respiration. The OCR indicated that irradiated adipocytes maintained a significantly reduced mitochondrial phosphorylation rate under both basal and stimulated conditions [Figure 8D]. To calculate the extracellular acidification rate (ECAR), glucose was added to the samples to induce the rate of glycolysis under basal conditions. Oligomycin was then added to block mitochondrial ATP production to increase ECAR to maximal glycolytic capacity. Finally, 2-DG (2-deoxy-glucose), a glucose analog, was added to inhibit glycolysis, resulting in a decrease in ECAR. The ECAR showed decreased glycolytic capacity in irradiated adipose tissue compared to unirradiated adipose tissue [Figure 8E]. Overall, radiation-exposed adipose tissue has a chronically altered metabolism that favors the release of free glycerols and free fatty acids and decreases glycolysis and mitochondrial oxidative phosphorylation.

### 2.9. Adipose Tissue Undergoes Senescence After Radiation Exposure

Another hallmark of dysfunctional adipose tissue is cellular senescence, or irreversible cell cycle arrest [47,48]. In adipose tissue, cellular senescence has been previously studied in the context of obesity and aging, but not after radiation exposure. In both cases, cellular senescence was associated with reduced adipogenesis and impaired adipose tissue function [46,47]. In senescent cells, p16 and p21, two cell cycle inhibitors, are highly upregulated. In irradiated adipose tissue, the expressions of p16 and p21 significantly increased 6 months after radiation exposure [Figure 9]. This increased cellular senescence provides a potential explanation for chronic radiation-induced adipose tissue dysfunction.

### 2.10. Immune Infiltration in Adipose Tissue Remains Heightened 6 Months After Radiation

Chronic immune infiltration is a common observation in obese adipose tissue. In this study, we previously noted the infiltration of pro-inflammatory immune cells in the adipose tissue of mice one month after radiation exposure [Figure 6]. To determine the duration of this infiltration, fat pads of pelvically irradiated mice were collected six months after radiation, and tissue sections were stained for the presence of CD4+ helper T-cells, CD8+ cytotoxic T-cells, and total macrophages. Similar to the results observed one month post-radiation, fat pads of irradiated mice at 6 months post-radiation contained significantly more CD4+ cells [Figure 10A] as compared to control tissues. However, there was no significant difference in CD8+ T-cells [Figure 10B]. Total macrophages remained significantly elevated in the fat of irradiated mice six months after pelvic irradiation [Figure 10C].

## 3. Discussion

The clinical use of radiation to treat cancer remains a common option for many cancers at various stages and locations in the body. Healthy tissues in the region of the tumor are often overlooked and not spared from radiation exposure. In healthy tissues, radiation causes elevated ROS, leading to chronic oxidative stress, inflammation, and fibrosis [49,50]. Additionally, individuals with adipose tissue dysfunction often have insulin resistance, inflammation, and hyperlipidemia. Adipose tissue dysregulation has been linked to persistent metabolic disorders and even cancer recurrence and metastasis [51,52]. Therefore, it is vital to understand the effects of radiation exposure on adipose tissue.

To the best of our knowledge, this is the first study to show that clinical dosing of radiation causes significant damage to adipose tissues. In the current study, we showed that irradiated adipose tissue is in a chronic state of dysregulation. Specifically, in vitro, irradiated fat is more oxidatively stressed, has increased lipolytic function, and is more senescent than unirradiated adipocytes. To create a clinically relevant model of the difference between irradiated and unirradiated adipose tissue in vivo, we exposed male C57BL/6 mice to 7.5 Gy of radiation to either the gonadal fat pads or the whole pelvis. We collected adipose tissues 1, 2, and 6 months post-radiation to assess radiation damage at acute and chronic timepoints. In this system, increased levels of oxidative stress, metabolic dysfunction, senescence, and inflammation were observed in irradiated adipose tissues for up to 6 months after radiation exposure. This suggests that radiation-induced injury to the adipose tissue can cause long-term damage. It is worth noting that adipose tissues become dysregulated during aging [53], potentially accounting for the elevated expression of oxidative stress markers and immune infiltration in the unirradiated control mice at the 6-month post-radiation timepoint.

Normal tissues exposed to radiation therapy often become oxidatively stressed because of the large increase in ROS [54]. The large accumulation of ROS leads to the disruption of redox homeostasis, which ultimately dysregulates typical cellular functions [55]. Using a combination of immunofluorescence and flow cytometry techniques, we observed that irradiated adipose tissue showed signs of oxidative damage through increased DNA/RNA oxidation and lipid peroxide levels. This corroborates the finding that unless protected from a sudden increase in ROS, healthy tissues are often damaged by radiation [56].

Chronic inflammation, a characteristic of damaged adipose tissue, is observed in irradiated adipose tissue. Adipose tissues have baseline levels of resident macrophages and T-cells [57], and animal models have revealed that whole-body radiation increases macrophage accumulation in adipose tissues [58]. Our data expanded these findings by showing that radiation to adipose tissue alone was enough to increase macrophage and T-cell accumulation and that this accumulation persisted 6 months after radiation treatment. RNA sequencing of adipose tissue 2 months after radiation revealed that inflammatory-related genes remained more highly expressed than in unirradiated adipose tissues. Macrophages can be characterized as either anti-inflammatory (M1) or pro-inflammatory (M2). Radiation can induce phenotype switching in macrophages, typically causing an increase in M2 macrophages [59]. In the present study, our data did not indicate whether the immune cells were pro-inflammatory or anti-inflammatory. However, inflammatory macrophage polarization plays a role in metabolic dysfunction by increasing lipolytic rates [60]. We were also unable to discern between resident and infiltrating macrophages, so we currently do not know the source of macrophages, which will be addressed in future studies.

We observed increased levels of lipolytic activity in irradiated adipocytes both in vitro and in vivo by measuring the activity of lipolytic enzymes, assessing the adipose tissue mass of the mice, and quantifying serum free fatty acids and free glycerol. In obesity, adipose tissue is highly oxidatively stressed. Elevated ROS levels are associated with metabolic dysfunction in obese mice [61]. Moreover, adult pediatric cancer survivors previously treated with abdominal radiation exhibited higher levels of adipose tissue dysfunction and were more likely to suffer from metabolic disorders than survivors not treated with radiation [62]. These data support the idea that upon adipose radiation, ROS increases and enhances the presence of inflammatory immune populations, resulting in chronic metabolic dysfunction.

Cellular senescence is believed to be a result of the DNA damage response and is a hallmark of radiation-induced damage. To date, no studies have investigated senescence in adipose tissue as a result of radiation exposure. In adipose tissues, aging and obesity are common inducers of senescence, as evaluated by an increase in β-gal staining and senescence-associated gene expression [63,64]. In our data, we have successfully shown that p16 and p21, markers of senescence, are both upregulated at the RNA level in irradiated adipose tissues and that irradiated adipose tissues have an increased number of β-gal^+^ cells compared to unirradiated adipose tissues.

This study has some limitations. In vitro, 3T3-L1 stem cells were used to create adipocytes; primary human adipocytes will be needed to investigate how radiation affects adipose tissue, as there are subtle differences between murine and human adipocytes [65]. In vivo, we used only male mice in our experiments. Experiments using female mice should be conducted to determine whether these changes are observed in the adipose tissue of female mice. Our study focused only on investigating the impact of radiation on visceral adipose tissue isolated from the gonadal fat pads of mice. Further studies are needed to determine the impact of clinically relevant radiation dosing on subcutaneous adipose tissue and other visceral adipose tissue depots. Lastly, this is an observational study, so we did not investigate the mechanism by which radiation damages adipose tissue. We previously showed that radiation reduces adiponectin (APN) expression [15]. As adipose tissues are the main producers of APN, we speculate that APN reduction could cause some of the observed damage to adipose tissues, but further experimentation is needed.

Radiation-induced tissue damage is a common occurrence in tissues exposed to radiation. However, unlike many other tissues, adipose tissue is atrophied and grossly abnormal following radiation exposure. Radiation-induced oxidative stress is a main culprit for much of the damage associated with radiation and contributes to chronic tissue damage [66]. Adipose tissue has a very high lipid content that is vulnerable to oxidation. Following radiation, adipose tissue maintains high levels of lipid peroxyl radicals, which cause irreversible damage to lipids. The oxidized lipid products then serve as secondary messengers, resulting in adduct formation on other cellular components, including DNA and proteins. Furthermore, fully differentiated cells, like mature adipocytes, decrease the DNA repair of radiation-induced double-strand breaks [67,68]. Taken together, we speculate that the increased lipid peroxidation occurrence and reduced ability to repair DNA damage result in adipose tissue being highly susceptible to radiation damage.

Many of the damages identified in irradiated adipose tissues are similar to those in obese adipose tissues, including oxidative stress, metabolic dysfunction, inflammation, and senescence. Obesity is a well-known risk factor for cancer development and poor survival [69,70]. This pairs well with the notable findings that radiation increases the risk of cancer recurrence and metastasis in prostate and breast cancer patients [17,18]. The work done in this study emphasizes the need for a greater understanding of adipose tissue health in relation to cancer and other diseases and strongly recommends that adipose tissue be protected from clinically prescribed radiation exposure in cancer patients.

## 4. Materials and Methods

### 4.1. Cell Culture and In Vitro Radiation

3T3-L1 mouse embryonic stem cells were obtained from ATCC (Manassas, VA, USA, CL-173). The cells were maintained in Dulbecco’s Modified Eagle Medium (DMEM) containing 10% fetal bovine serum (FBS) and 1% penicillin/streptomycin. Differentiation of 3T3-L1 cells into adipocytes was accomplished by treating the cells with a differentiation cocktail of 1 µM dexamethasone, 10 µg/mL insulin, 0.5 mM 3-isobutyl-1-methylxanthine (IBMX), and DMEM containing 10% FBS and 1% penicillin/streptomycin. After 48 h, the differentiation medium was replaced with maintenance media, DMEM containing 10 μg/mL insulin, 10% FBS, and 1% penicillin/streptomycin. The medium was changed every two days. After 10 days, the cells were fully differentiated and contained large lipid droplets, as determined by Oil Red O staining. Starting on day 10, 3 Gy of radiation was administered daily for 3 days to the adipocytes using a RadSource 2000 X-Ray Box Irradiator (RadSource, Buford, GA, USA) at 1.2 Gy/min. The cells received a total dose of 9 Gy of radiation. 

### 4.2. Experimental Animals

C57Bl/6 male mice (aged 6–8 weeks) were obtained from Jackson Laboratories. They were housed at the University of Nebraska Medical Center (UNMC) in accordance with the Guide for Care and Use of Laboratory Animals by the National Institutes of Health. The mice were kept on a 12 h light/dark cycle and fed and watered ad libitum. Animal treatment procedures were approved by the UNMC Institutional Animal Care and Use Committee (20-019-03-FC).

### 4.3. Animal Radiation Treatments

For 1 month post-radiation exposure experiments, using the image-guided Small Animal Research Radiation Platform (SARRP, Xstrahl, Suwanee, GA, USA), 7.5 Gy of radiation (3 Gy/min) for 5 days (37.5 Gy total) was directed through a 5 × 5 mm collimator at the gonadal fat pads of C57Bl/6 mice, aged 8–10 weeks using pre-radiation planning with CT imaging.

For the 2 and 6 month post-radiation exposure experiments, C57Bl/6 mice (6–8 weeks old) were anesthetized with an xylazine (11 mg/kg) and ketamine (80 mg/kg) solution intraperitoneally (ip). Their upper bodies were lead-shielded, exposing the pelvis to X-ray irradiation at 1.2 Gy/min. The mice received 7.5 Gy for five consecutive days using a RadSource 2000 X-Ray Box Irradiator. The irradiated pelvic area measured approximately 2 × 2 cm^2^.

### 4.4. Immunofluorescence

Immunofluorescence staining of the 3T3-L1 adipocytes was performed as follows. The cells were differentiated and irradiated with 3 Gy for three consecutive days. At the appropriate time points, the cells were trypsinized and cytospun onto glass slides (10,000 cells/slide). The cells were fixed on slides using 4% paraformaldehyde for 15 min and washed three times with PBS. Immunofluorescence staining of the tissue sections was performed as previously described [15]. For paraffin-embedded tissues, the slides were deparaffinized in xylenes and rehydrated in graded alcohols. For antigen retrieval, the slides were heated to 95 °C in 0.01 M sodium citrate buffer (pH 6.0) with 0.05% Tween 20. The slides were washed twice in PBS for 5 min and then blocked with 10% goat serum for 1 h at room temperature. Primary antibodies were diluted in goat serum and incubated at 4 °C overnight. The slides were washed three times with PBS and incubated with fluorescent secondary antibodies for 1 h at RT in the dark. The cells were mounted with Prolong Gold Antifade DAPI with mounting solution and imaged using a Leica DM4000 B LED microscope (Leica, Deerfield, IL, USA).

Antibodies used: Oxidative stress was measured using the monoclonal primary antibodies mouse anti-DNA/RNA Damage (Abcam, Cambridge, UK, AB15A3, 1:800) and stained with secondary goat anti-mouse AlexaFluor 488 (Invitrogen, Waltham, MA, USA, A-28175; 1:500) as well as mouse anti-4-hydroxynoneonal (ThermoFisher, Waltham, MA, USA, BS-6313R, 1:500) stained with secondary goat anti-mouse AlexaFluor 647 (Invitrogen A-21235; 1:500) to measure 8-OHdG and lipid peroxidation, respectively. To minimize non-specific staining, a Mouse-on-Mouse kit (Vector Laboratories, Newark, CA, USA, FMK-2201) was used according to the manufacturer’s recommendations. After the images were taken, their mean fluorescence intensity was quantified using ImageJ software version 1.48 and normalized to the control.

Immune infiltration was quantified by staining the tissue slides with either rabbit monoclonal primary anti-CD4 (Abcam 183685; 1:1000), primary rabbit monoclonal anti-CD8 (Abcam 217344; 1:2000), or primary rabbit monoclonal anti-F4/80 (Abcam 111101; 1:50), followed by staining with secondary goat anti-rabbit AlexaFluor 488 (Invitrogen A-11008; 1:500). ImageJ was used to quantify the images. Cells with high fluorescence staining were considered positive and reported as the mean number of positive cells per image per mouse.

Cellular senescence was observed by staining the tissue slides with rabbit monoclonal primary anti-p16 INK4A (Cell Signaling, Danvers, MA, USA, 18769; 1:200) and secondary goat anti-rabbit AlexaFluor 647 (Invitrogen A-21245; 1:500). Rabbit monoclonal primary anti-p21 (Abcam AB109199, 1:250) and secondary goat anti-rabbit AlexaFluor 488 (Invitrogen A-11006; 1:500) were used to visualize p21 expression on the tissue slides. ImageJ software was used to determine high fluorescence staining, indicating cells positive for p16 and p21. DAPI was used to enumerate the total number of cells per image. Both senescence markers are reported as the percentage of positive cells per image.

### 4.5. Oxidative Stress Markers Evaluated via Flow Cytometry

Total reactive oxygen species (ROS), total superoxide, mitochondrial superoxide, and lipid peroxidation were detected using flow cytometry. 3T3-L1 adipocytes were seeded at a density of 3 × 10^5^ cells/well. After differentiation and irradiation (3 Gy × 3 days or sham), the cells were collected 48 h post-radiation. To measure total ROS and superoxide, the cells were detached, washed in Hanks’ balanced salt solution (HBSS), and incubated with dihydroethidium (DHE, ThermoFisher, 5 μM) for 20 min at 37 °C in the dark. The cells were subjected to flow cytometric analysis using an LSRII Green 532 Flow Cytometer (BD Biosciences, Franklin Lakes, NJ, USA). Superoxide was measured using 405/570 nm excitation/emission wavelengths. MitoSOX was used to measure mitochondrial superoxide levels. Briefly, the cells were detached, washed with HBSS, and incubated in the dark with MitoSOX Red (ThermoFisher, 5 μM) for 10 min at 37 °C. Following incubation, the cells were washed twice with HBSS before analysis. Mitochondrial superoxide levels were measured with an LSRII Green 532 Flow Cytometer using a 585/42 nm band pass with a 550 nm long-pass emission filter. Lipid peroxidation was measured by incubating cells with BODIPY 581/591 C11 (ThermoFisher, 2 μM) for 30 min in the dark at 37 °C. The cells were then collected and washed with HBSS. Lipid peroxidation was quantified with an LSRII Green 532 Flow Cytometer, BD Biosciences, to detect FITC (530/30) and PE (582/15). FACSDiVa analysis software (BD Biosciences, version 9.0) was used to analyze the data.

### 4.6. Cell Viability

3T3-L1 cells were cultured and differentiated, as stated in Section 4.1. The cellular viability of irradiated and control adipocytes was quantified 48 h following the last dose of radiation using trypan blue exclusion. Briefly, the medium was collected and adipocytes were detached with a cell scraper, centrifuged, and resuspended in 2.5 mL of fresh media. Cell suspension (200 μL) was loaded on a Vi-CELL BLU (Beckman Coulter, Brea, CA, USA) automated cell viability analyzer, per the manufacturer’s instructions. Data is presented as % viability.

### 4.7. Intracellular Lipid Staining

BODIPY: Twenty-four hours before radiation, 3T3-L1 adipocytes were stained with 10 μM BODIPY 630/650 NHS Ester (Invitrogen D10000) for 2 h before washing with PBS to remove excess stain. The cells were exposed to 3 Gy of radiation for 3 days and then cytospun onto glass slides 48 h after radiation. DAPI was used as the counterstain. Slides were imaged using a Leica DM4000 B LED microscope. ImageJ software was used to quantify the mean fluorescence intensity of the BODIPY staining.

Oil Red O: Forty-eight hours after radiation, 3T3-L1 adipocytes were fixed in 4% paraformaldehyde for 15 min. After fixation, the cells were washed with sterile water and incubated in 60% isopropanol at room temperature for 2 min. Isopropanol was removed, and Oil Red O stain (Sigma-Aldrich, St. Louis, MO, USA, 01391, 3:2 ratio with DI water) was added for 5 min to cover the cells completely. The Oil Red R stain was rinsed with tap water until the water was clear. The plates were kept in water until they were ready to be viewed. Brightfield images were captured with an Olympus IX81 inverted microscope (Olympus, Center Valley, PA, USA). The lipid droplet size was quantified by tracing the red border of Oil Red O-stained lipid droplets using ImageJ software.

### 4.8. Lipolytic Activity Methodology

Twenty-four hours before radiation, fully differentiated 3T3-L1 adipocytes were stained with 10 μM BODIPY FL C16 (Invitrogen, D3821) for 2 h, before washing with PBS and replacing the media with fresh maintenance media. Immediately following the last dose of radiation, the medium was replaced with fresh maintenance medium. Twenty-four hours after radiation, the conditioned medium from adipocytes was collected. Fluorescence intensity was measured with an excitation wavelength of 505 nm and an emission of 551 nm using an Infinite M200 Pro plate reader (Tecan, Mannedorf, Switzerland).

Additionally, 3T3-L1 adipocytes were differentiated and irradiated as described above and lysed 24 and 48 h after radiation. The lipolytic activity of adipocytes was assessed using a fluorometric-based Lipase Activity Assay Kit (Cayman Chemical, Ann Arbor, MI, USA, 700640). The cells were collected and treated according to the manufacturer’s protocol. Briefly, the lysed samples were diluted to 1:4 in 1x Assay Buffer. In a solid white 96-well plate, the diluted samples (10 µL) were mixed with 170 uL of 1x Assay Buffer and 10 µL of a Thiol Detector. To initiate the lipase reaction, 10 µL Lipase Substrate was added to the sample wells. Fluorescence was measured by an Infinite M200 Pro plate reader (Tecan) at 37 °C every 30 s for 15 min at an excitation wavelength of 380 nm and an emission wavelength of 510 nm. Lipase activity was analyzed as the change in fluorescence (RFU) per min.

### 4.9. Immunoblotting

Western blotting was performed as previously described [71]. Briefly, 3T3-L1 adipocytes were collected, and cell pellets were lysed in cell lysis buffer [120 mM NaCl, 50 mM Tris-HCl, 5 mM EDTA, 1% NP-40, and complete protease inhibitor cocktail tablets (Roche, Basel, Switzerland, cat # 11697498001; 1 tablet/50 mL)] to prepare whole-cell extracts. The samples were electrophoresed on Bolt 4–12% Bis-Tris Plus gels (ThermoFisher, cat # NW04120BOX) and transferred to nitrocellulose membranes (Life Technologies, Carlsbad, CA, USA, cat # IB23002). The membranes were blocked for 1 h with nonfat milk (5%) in Tris-buffered saline containing 0.5% Tween 20 (TBST) and incubated overnight with primary antibodies. The membranes were washed with TBST and incubated with horseradish peroxidase (HRP)-conjugated secondary antibodies (1:10,000 dilution) for 1–2 h at room temperature. Blots were developed using an ECL detection system (ThermoFisher Scientific) and exposed to X-ray film after washing with TBST. ImageJ software was used for densitometry analysis.

Antibodies used: Fatty acid synthase (FASN) was quantified using a primary polyclonal antibody against FASN (ProteinTech, Sankt Leon-Rot, Germany, 66591; 1:20,000) and stained with anti-rabbit IgG, HRP-linked secondary antibody (Cell Signaling 7074S; 1:10,000). Lipolysis was quantified using the ratio of primary phosphorylated (Ser565) hormone-sensitive lipase (HSL) to total HSL (Cell Signaling 4137 and 4107; 1:1000), and anti-rabbit IgG, HRP-linked secondary antibody (Cell Signaling 7074S; 1:10,000). The senescence marker used was the primary monoclonal antibody rabbit anti-p21 (Abcam AB109199, 1:1000) stained with anti-rabbit IgG, HRP-linked secondary antibody (Cell Signaling 7074S; 1:10,000).

### 4.10. RNA Sequencing

The mice were irradiated as described above, receiving 7.5 Gy for 5 days. Primary mouse gonadal adipose tissue was harvested two months after radiation. Fat pads were digested with collagenase and incubated at 37 °C for 45 min, with manual swirling every 5 min. Once the fat was dissociated, a 4X volume of cold HBSS (without Ca^2+^ & Mg^2+^) was added, and the tube was centrifuged at 450 g for 3 min at 4 °C. Mature adipocytes found in the floating fraction were removed and cultured for 3 days. Total cellular RNA was isolated from primary adipocytes using the Quick-RNA™ MiniPrep (Plus) kit from Zymo Research, Irvine, CA, USA, following the manufacturer’s protocol after genomic DNA removal and DNase treatment. An Infinite M200 Pro plate reader (Tecan) was used to measure the concentration and quality of total RNA. Total RNAs with a 260/280 ratio of ~2.0 were used for further experiments. Total RNA was sent to City of Hope, CA, for RNA sequencing. Sample quality control was performed using the Bioanalyzer RNA6000 Nano ChIP. The sequencing library was prepared using the Illumina Stranded RNA-seq library preparation protocol. Quality control of the sequencing library was ensured using Qubit. RNA sequencing was performed using the RNA-seq Ribozero-stranded protocol, which removes ribosomal RNA to enhance the mRNA signal in the HiSeq 2500 Fast mode SE50.

### 4.11. Quantitative RT-PCR

Seven days post-radiation, total RNA from 3T3 cells was isolated using a Qiagen (Venlo, Netherlands) RNA isolation kit. The purity of the isolated RNA was analyzed using a Tecan Infinite 200Pro (Ref. 30050303) nanodrop. RNA (1 ug) was used for cDNA synthesis using Applied Biosystems (Ref. 4368814). For quantitative real-time PCR, SYBR Green Master Mix (Roche) was used, and the cycle conditions were 5 min at 95 °C, 95 °C for 30 s, 1 min at 58 °C, and 30 s at 72 °C for 39 cycles. The expression levels were normalized to RPLP0, and the data are represented as fold changes compared to the control conditions. The mouse primer sequences used in this study were p16 (F-5′-CCCAACGCCCCGAACT-3′; R-5′-GCAGAAGAGCTGCTACGTGAA-3′), p21 (F-5′-CCCGCCTTTTTCTTCTTAGC-3′; R-5′-TTCTCATGCCATTCCTTTCC-3′), and RPLP0 (F-5′-GCAGGTGTTTGACAACGGCAG-3′; R-5′-GATGATGGAGTGTGGCACCGA-3′).

### 4.12. β-Galactosidase Staining

A detailed senescence staining procedure was performed as previously described [72]. Briefly, seven days post-radiation, the cells were fixed with 4% paraformaldehyde for 3 min. After washing twice with PBS, the cells were incubated with senescence-associated β-galactosidase (SA-β-Gal) staining solution (0.1% X-gal, 5 mM potassium ferrocyanide, 5 mM potassium ferricyanide, 150 mM sodium chloride, and 2 mM magnesium chloride in 40 mM citric acid/sodium phosphate solution, pH 6.0 [73]) for 48 h at 37 °C to avoid light exposure. Brightfield images were captured with an Olympus IX81 inverted microscope. Cells exhibiting positive β-Gal staining (green stain) were quantified, and the data are presented as β-Gal^+^ cells per image.

### 4.13. Estimation of Adipocyte Size

Adipocyte size was quantified using hematoxylin and eosin (H&E) stained tissue slides collected from mice one month after fat pad radiation. Brightfield images were obtained using a Leica DM4000 B LED microscope. ImageJ software (version 1.48) was used to measure the area of the adipocytes by tracing the borders of the cells. Adipocytes in five fields per mouse were traced and quantified for each treatment group.

### 4.14. Free Fatty Acid and Free Glycerol Assays

The 3T3 adipocytes were irradiated with 3 Gy for 3 days. Seventy-two hours post-radiation, free fatty acids (Abcam 65341) from the conditioned media were measured per the manufacturer’s instructions. For the in vivo studies, blood was collected from the thoracic cavity at harvest, 6 months post-radiation, immediately after euthanasia by severing the aorta. The collected blood was centrifuged at 3000× *g* for 10 min, and the serum was extracted and prepared for assaying or immediately frozen and stored at −20 °C. Serum free fatty acids (Abcam 65341) and free glycerol (Abcam 65337) levels were measured according to the manufacturer’s instructions.

### 4.15. Seahorse Analysis

The mice were treated with pelvic radiation, as previously described. Six months later, primary adipocytes were isolated from the gonadal fat pads. Fat pads were digested with collagenase and incubated at 37 °C for 45 min, with manual swirling every 5 min. Once the fat was dissociated, a 4X volume of cold HBSS (without Ca^2+^ & Mg ^2+^) was added, and the tube was centrifuged at 450× *g* for 3 min at 4 °C. Mature adipocytes found in the floating fraction were removed and cultured for 3 days. The oxygen consumption rate and extracellular acidification of primary adipocytes were assessed using an Agilent Seahorse XF analyzer, as described previously [74].

### 4.16. Statistical Analysis

GraphPad Prism v9 was used for all statistical analyses. Mean and standard deviation values from three independent experiments were used for statistical analysis of all in vitro experiments performed. For ex vivo and in vivo experiments, each group consisted of 5–10 animals. All data is represented as the mean ± standard deviation (SD). Unless otherwise described, significant differences between the groups were determined using Student’s *t*-test, and a *p*-value ≤ 0.05 was considered statistically significant.

## Figures and Tables

**Figure 1 ijms-26-05626-f001:**
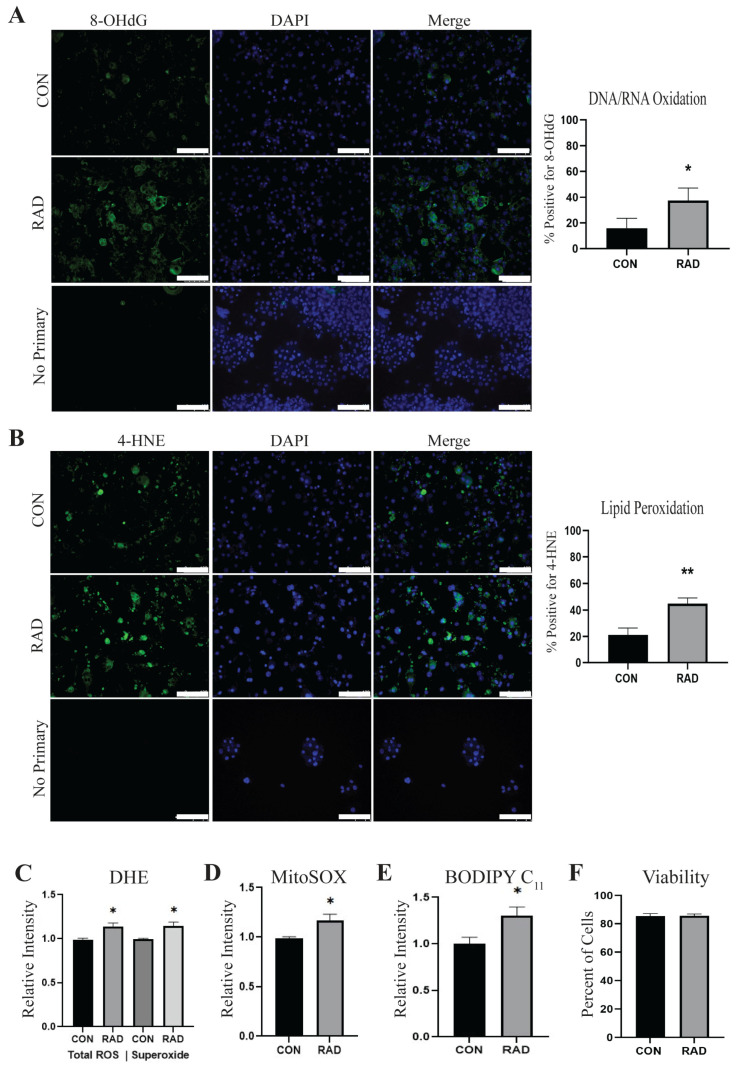
**Radiation causes oxidative stress in 3T3-L1 adipocytes.** 3T3-L1 adipocytes were irradiated with 0 Gy (CON) or 3 Gy for 3 days (RAD). (**A**) Two days after radiation, adipocytes were fixed and stained for 8-OHdG, a marker of DNA/RNA oxidation (green) or DAPI (blue), and (**B**) 4-HNE, an indication of lipid peroxidation (green). (**C**) Total ROS and superoxide levels were quantified in adipocytes 48 h following radiation or sham via DHE staining. (**D**) MitoSOX staining was used to quantify mitochondrial superoxide, and (**E**) BODIPY C_11_ to quantify lipid peroxidation in CON and RAD adipocytes 48 h post-radiation. (**F**) Viability was not altered in adipocytes treated with radiation. Six random fields of view per independent replicate (*n* = 3) were averaged for quantification. The white bar represents 100 µm. * and ** indicate a significant difference (*p* ≤ 0.05 and *p* ≤ 0.01, respectively; *n* = 3) from the control.

**Figure 2 ijms-26-05626-f002:**
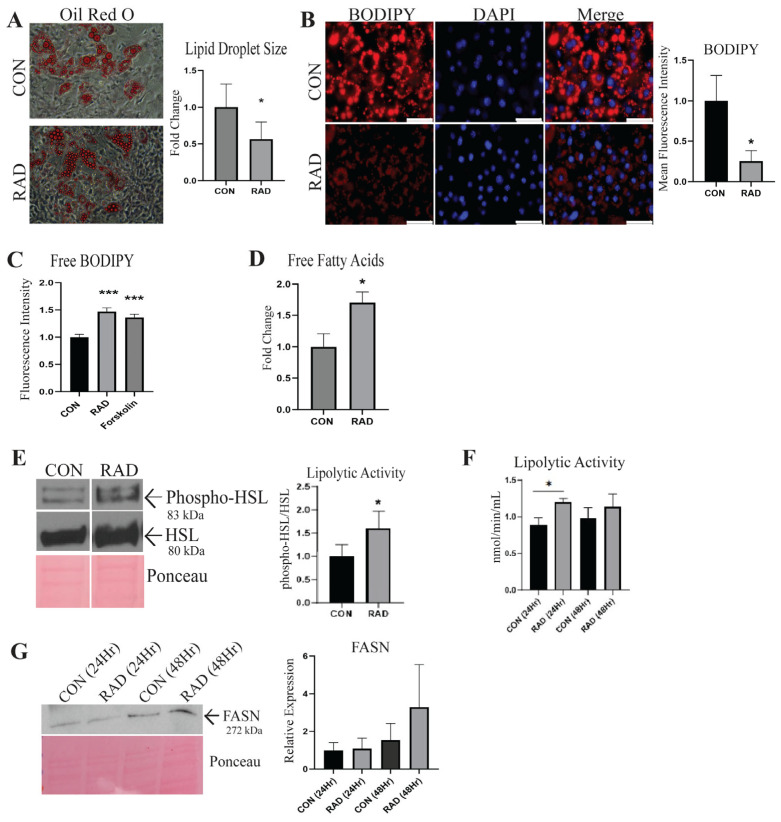
**Radiation causes metabolic dysfunction in adipocytes.** (**A**) Oil Red O was used to stain neutral lipids within adipocytes treated with 0 Gy (CON) or 3 Gy (RAD) and to quantify lipid droplet size (red color indicates lipids, average of 100 lipid droplets/image, 4 images/biological replicate, *n* = 3). Image was taken at 30× magnification. (**B**) BODIPY (red stain) was used to stain and quantify lipids within the CON and RAD adipocytes. (**C**) Quantification of BODIPY fluorescence in the medium of CON and RAD adipocytes 48 h post-radiation. The white bar represents 100 µm. (**D**) Quantification of free fatty acids found within the conditioned media of CON and RAD adipocytes 72 h post-radiation. (**E**) Representative western blots of phospho-HSL (Ser565), total HSL, and Ponceau, a loading control. Lipolytic activity was measured by normalizing the densitometry of phospho-HSL to total HSL. (**F**) Lipolytic activity of CON and RAD adipocytes 24 and 48 h after radiation. (**G**) Representative blots with loading control (Ponceau). FASN (normalized to Ponceau) was measured by western blotting in adipocytes 24 and 48 h after radiation. * and *** indicate a significant difference (*p* ≤ 0.05 and *p* ≤ 0.01, respectively; *n* = 3) from the control, as measured using Student’s *t*-test.

**Figure 3 ijms-26-05626-f003:**
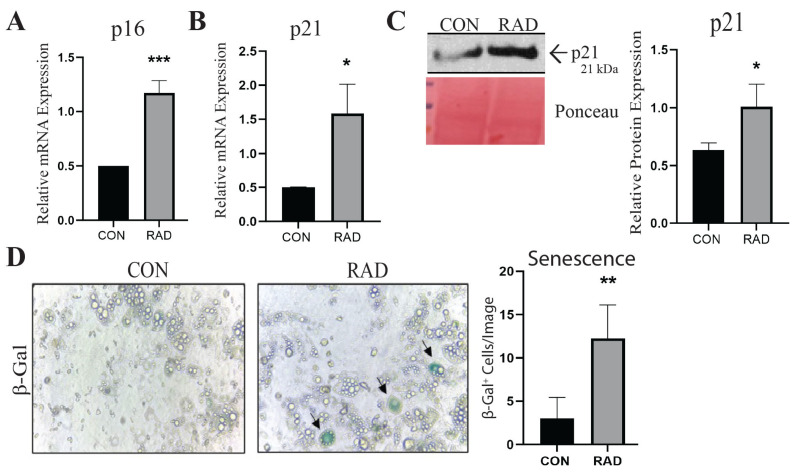
**Irradiated adipocytes undergo cellular senescence 7 days after radiation exposure.** (**A**) RT-PCR was used to measure the presence and relative abundance of p16 and (**B**) p21 mRNA expression in unirradiated (CON) and irradiated (RAD) and 3T3 adipocytes. (**C**) Representative western blots and quantification of p21 protein expression (normalized to ponceau, loading control) in CON and RAD adipocytes. (**D**) Representative images and quantification of β-Gal staining in CON and RAD adipocytes. Image was taken at 30× magnification. In the representative images, black arrows point toward β-Gal^+^ cells. *, **, and *** indicate significant differences (*p* ≤ 0.05, 0.01, and 0.001, respectively; *n* = 3) compared to the control, as measured using Student’s *t*-test.

**Figure 4 ijms-26-05626-f004:**
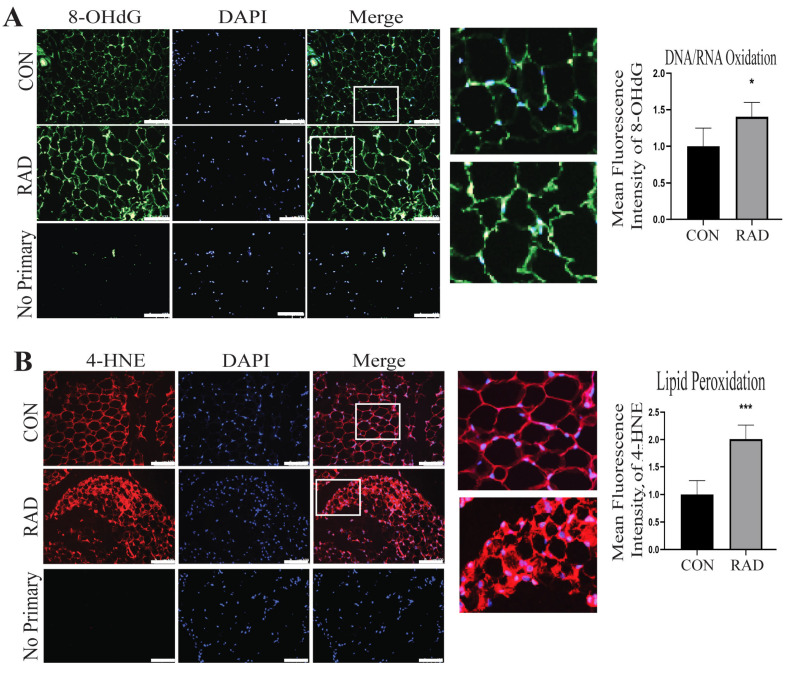
**In vivo, irradiated adipose tissue acutely sustains oxidative damage.** To assess radiation-induced oxidative damage in murine adipose tissue, mice were irradiated with 7.5 Gy of radiation for 5 days (RAD) or 0 Gy (CON). Adipose tissue was harvested one month post-radiation. (**A**) Adipose tissue sections were stained for 8-OHdG, an indicator of DNA/RNA oxidation (green color), and (**B**) 4-HNE, an indicator of lipid peroxidation (red color, 5 images/mouse, 5 mice/group). White box is the area that is magnified 10× and shown at the immediate right. * and *** indicates a significant difference (*p* ≤ 0.05 and *p* ≤ 0.001, respectively) compared to the control. The white bar represents a distance of 100 µm.

**Figure 5 ijms-26-05626-f005:**
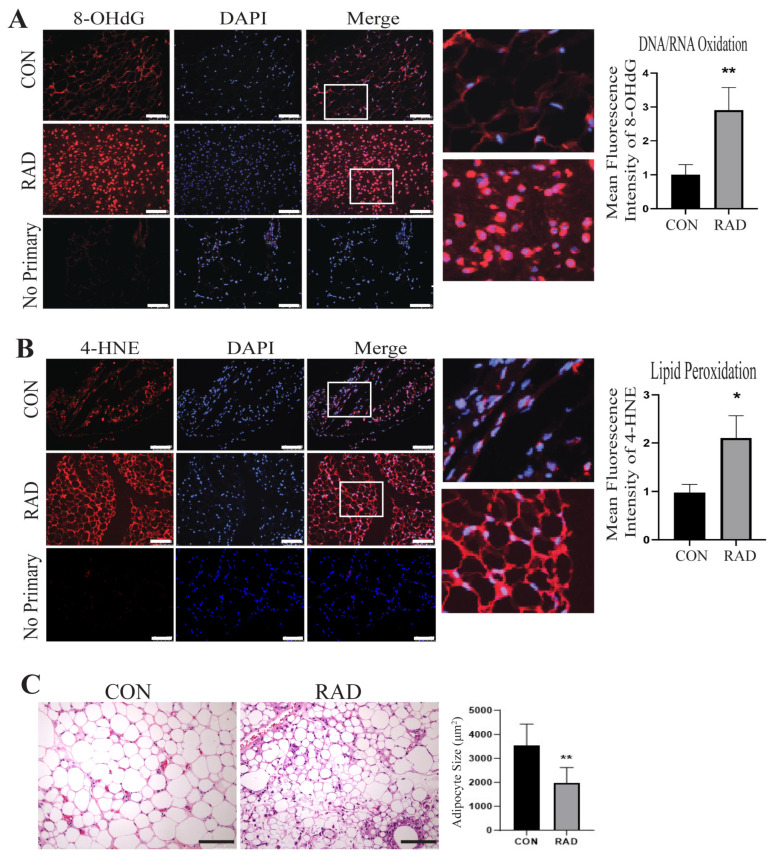
**Two months post-radiation, adipose tissues exhibit signs of oxidative damage.** To assess radiation-induced oxidative damage in murine adipose tissue, mice were irradiated with 7.5 Gy of radiation for 5 days (RAD) or 0 Gy (CON). Adipose tissue was harvested two months post-radiation. (**A**) Adipose tissue sections were stained for 8-OHdG, (red color) an indicator of DNA/RNA oxidation, DAPI is stained in blue, and (**B**) 4-HNE, an indicator of lipid peroxidation (red color) (5 images/mouse, 5 mice/group). The white bar represents a distance of 100 µm. White box is the area that is magnified 10× and shown at the immediate right. (**C**) Adipose tissue slides were stained with H&E. A total of 150 adipocytes/images were traced and size-quantified (5 images/mouse, 5 mice/group). The black bar represents 100 µm. * and ** indicates a significant difference (*p* ≤ 0.05 and *p* ≤ 0.01, respectively) compared to the control.

**Figure 6 ijms-26-05626-f006:**
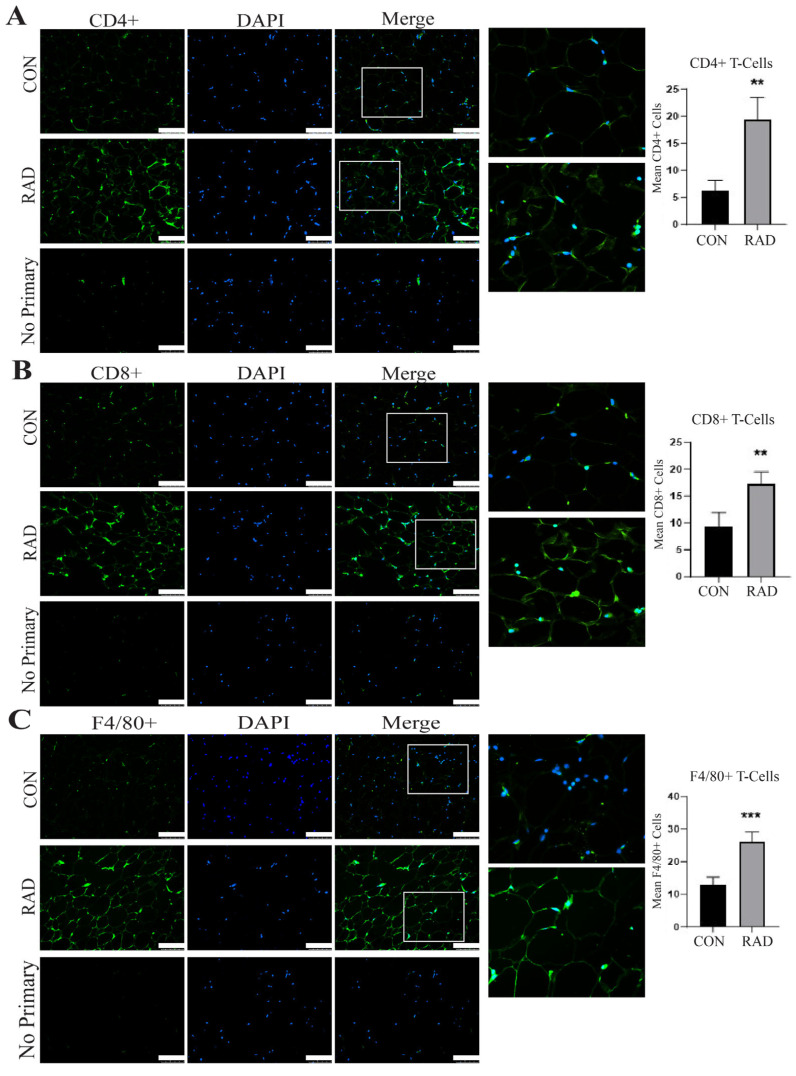
**Radiation increases immune infiltration acutely in murine adipose tissue.** Adipose tissue was collected one month after the mice were irradiated with 7.5 Gy × 5 days (RAD) or unirradiated (CON). Immunofluorescence staining was completed to quantify immune infiltration. (**A**) CD4+ T-cells (green staining) DAPI is blue stain, (**B**) CD8+ T-cells (green staining), and (**C**) F4/80+ cells (green staining) were quantified in 5 images per mouse and 5 mice per group. White box is the area that is magnified 10× and shown at the immediate right. The white bar represents a distance of 100 µm. ** and *** indicate a significant difference (*p* ≤ 0.01 and *p* ≤ 0.001, respectively) compared to the control.

**Figure 7 ijms-26-05626-f007:**
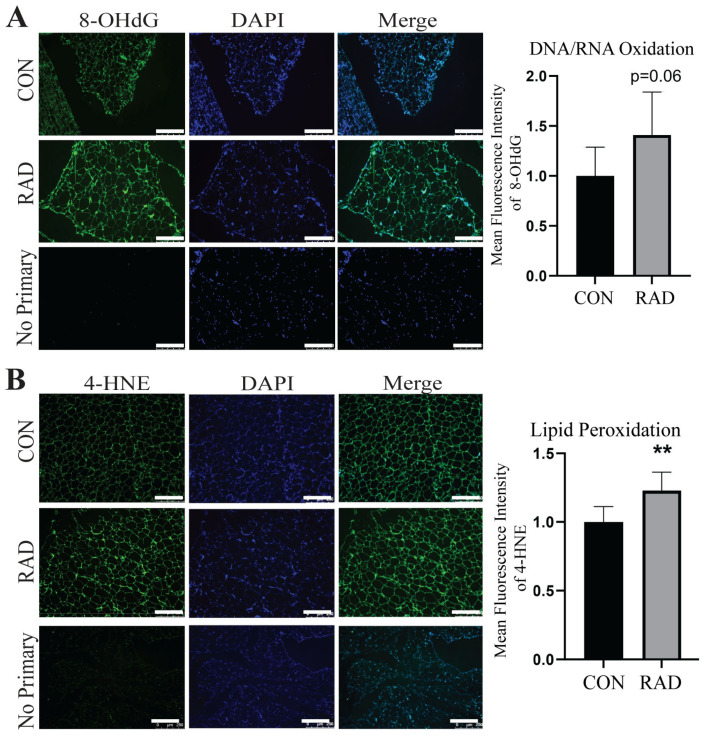
**Radiation causes chronic oxidative damage to adipose tissue.** The mice received 7.5 Gy of radiation for 5 days to the pelvis (RAD) and control (CON) 6 months post-radiation. (**A**) Representative images and adipose tissue sections stained with 8-OHdG, an indicator of DNA/RNA oxidative damage, and quantification (8-OHdG = green, DAPI = blue). (**B**) Representative images of adipose tissue sections stained with 4-HNE, an indicator of lipid peroxidation, and quantification (4-HNE = green, DAPI = blue). Six or more images were randomly collected and analyzed per mouse (*n* = 5 mice/group). The white bar indicates 250 µm. ** = *p* ≤ 0.01.

**Figure 8 ijms-26-05626-f008:**
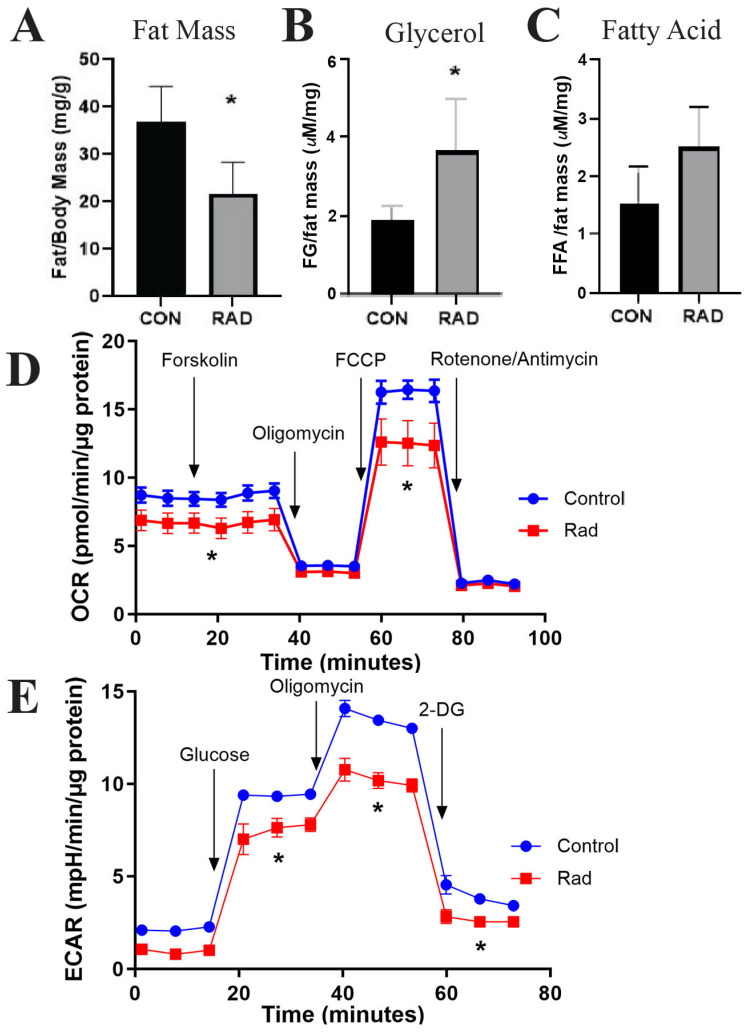
**Metabolic dysfunction is observed in irradiated adipose tissue.** The mice were exposed to 7.5 Gy of pelvic radiation for 5 days. (**A**) Six months after radiation, the mass of the adipose tissue (mg) was collected and normalized to body mass (g). (**B**) Free glycerol and (**C**) free fatty acids were measured and quantified from the mice’s serum. (**D**) Oxygen consumption rate (OCR) and (**E**) extracellular acidification rate (ECAR) were quantified via Seahorse from adipocytes isolated from control and irradiated murine adipose tissues. *n* = 5 mice/group and * indicates a significant difference (*p* ≤ 0.05) compared to the control.

**Figure 9 ijms-26-05626-f009:**
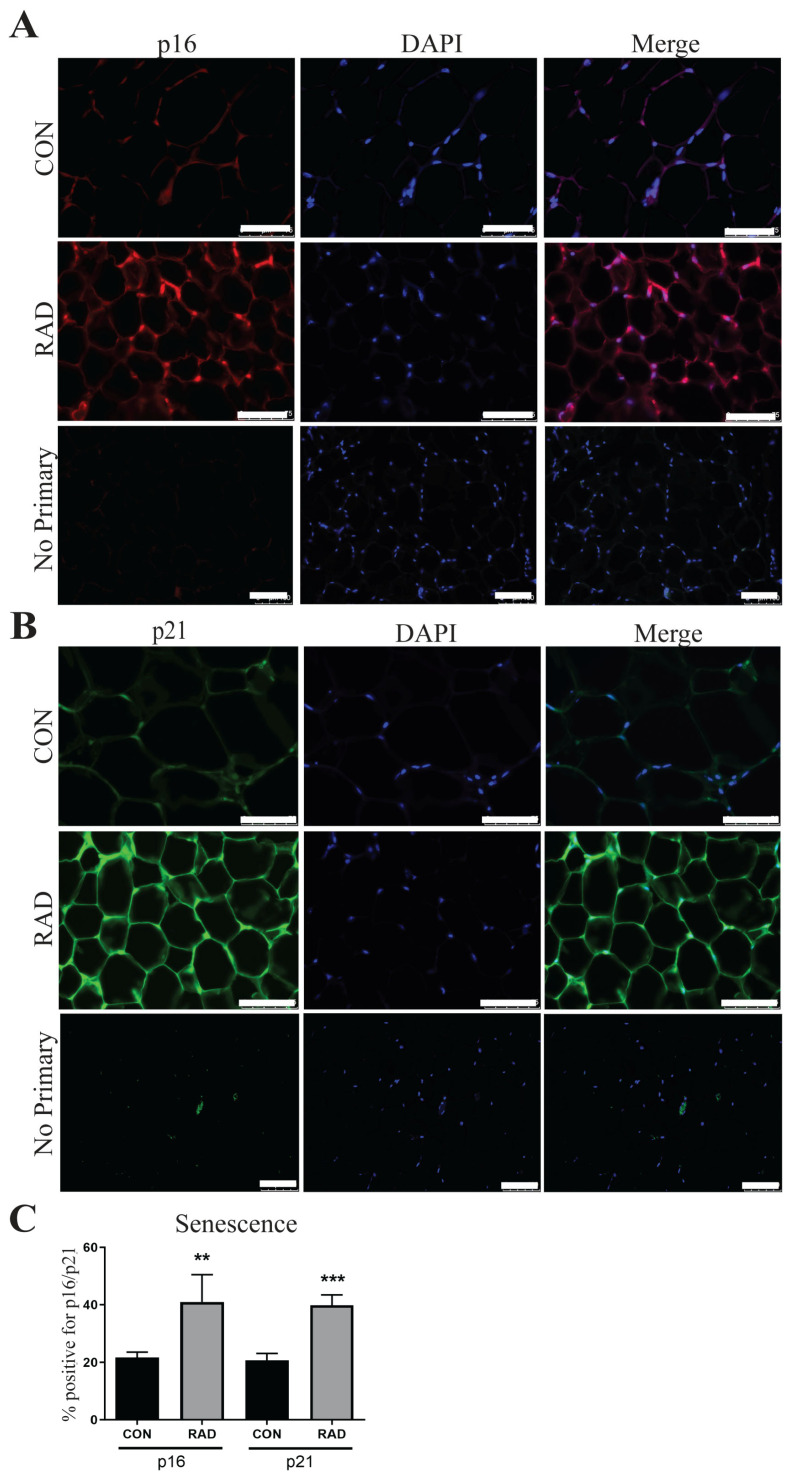
**Markers of senescence remain elevated in irradiated adipose tissue 6 months after radiation exposure.** (**A**) Representative images of adipose tissue sections from control and irradiated mice were collected and stained for p16, an indicator of senescence (p16 = red, DAPI = blue). (**B**) Adipose tissue sections were stained for p21, a senescence marker (p21 = green, DAPI = blue). (**C**) Quantification of p16 and p21. Each group represents a minimum of six fields/mouse, 5 mice/group. The white bar represents 100 µm. ** and *** indicates significant difference (*p* ≤ 0.01 and *p* ≤ 0.001) from the control.

**Figure 10 ijms-26-05626-f010:**
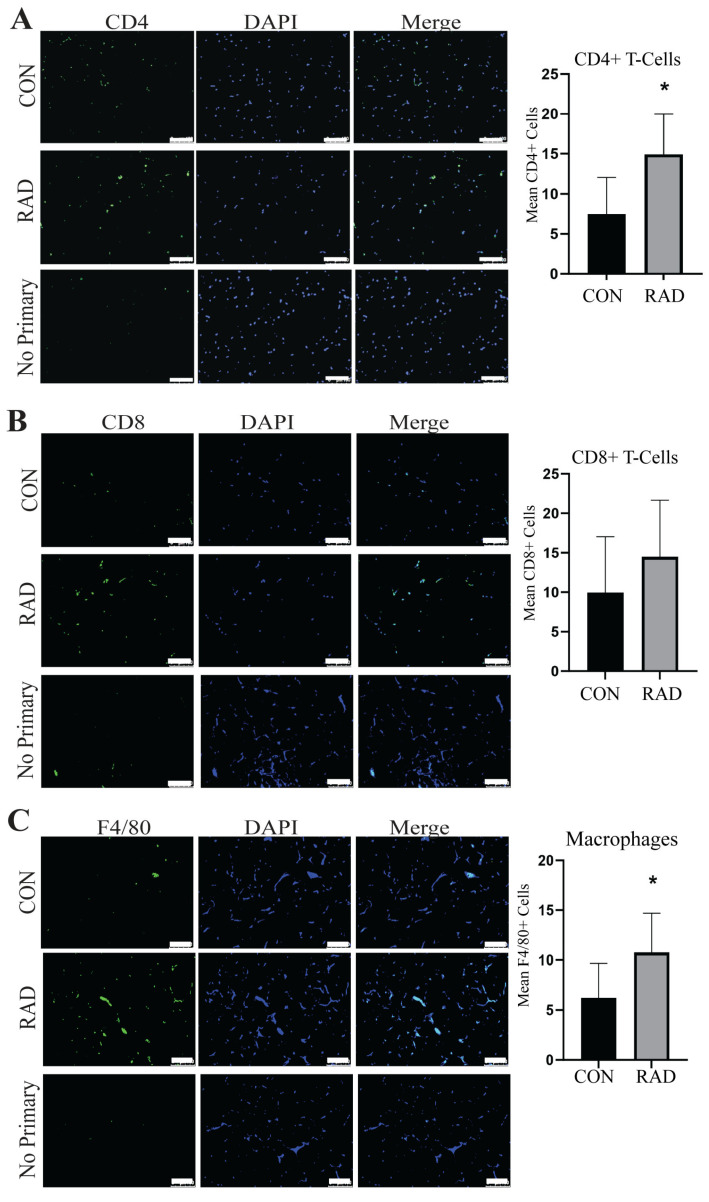
**Immune infiltration remains present in adipose tissue 6 months after radiation exposure, 7.5 Gy for 5 days.** (**A**) Representative images and quantification of adipose tissues stained for CD4+ T-cells (green staining) and DAPI blue staining in unirradiated (CON) and irradiated (RAD) adipose tissue sections collected from the mice 6 months after exposure to pelvic radiation. (**B**) CD8+ T-cell (green staining) and (**C**) F4/80+ macrophage (green staining) in CON and RAD adipose tissue 6 months post-radiation. A minimum of six fields/mouse were analyzed at random from the positive staining (≥6 fields/mouse, 8 mice/group). The white bar represents 100 µm and * indicates a significant difference (*p* ≤ 0.05) from the control.

## Data Availability

The RNA sequencing data that support the findings of this study are openly available in NCBI’s Gene Expression Omnibus and are accessible through GEO series accession number GSE295098 on 29 May 2025 (https://www.ncbi.nlm.nih.gov/geo/query/acc.cgi?acc=GSE295098).

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
