# Peer review of "Radiation Promotes Acute and Chronic Damage to Adipose Tissue"

_ijms, 2025, doi:10.3390/ijms26125626_

Round 1
Reviewer 1 Report
Comments and Suggestions for Authors
The manuscript characterizes radiation effects on adipocytes in culture and adipose tissue from a mouse model. Characterization focuses on oxidative stress, metabolic dysfunction, immune response, and senescence. Acute (few days) and chronic (months) radiation effects are investigated. Fluorescence-based sensing and immunostaining are the primary means of characterization.
Overall, the manuscript appears technically sound and is well documented.
The following are concerns and comments.
- Because there are many materials and methods are used in the manuscript, it would be helpful to explicitly state at the start of each Results subsection or in each figure caption where the methods are found (i.e., which subsection(s) of the Materials and Methods section).
- In section 2.1, how is viability being quantified?
- Figure 1 – what does “2° only” mean?
- Figure 1 - “A minimum of six fields/biological replicates (n=3) were averaged for quantification.” Does this mean six (6) fields of view each for three (3) independently prepared samples? Please rephrase for clarity. Is this true for all figures? How were fields of view chosen?
- Why are DAPI images shown in the second column in some figures (e.g., Figure 1) and first column in other images (e.g., Figure 5)? If there is no reason, please be consistent. It will help the reader navigate the many figures.
- The meaning of *** is missing in the caption for Figure 3. (I did not check all figures.) If the meaning of *, **, and *** is the same for all figures, say so.
- P11, line 248-250 – “In mice that received pelvic radiation, serum free glycerol [Figure 7B] and free fatty acids [Figure 7C] were elevated compared with the serum of unirradiated mice; however, only free glycerol levels were significantly elevated.” This sentence is confusing, the second part in particular. Why is glycerol mentioned twice? Are you concluding that levels are elevated or not?
- Figure 7 D, E – many functional modifiers are added to the sample. Please remind the reader of their effects, e.g., oligomycin decreases OCR by blocking ATP synthase proton channel, FCCP increases OCR by uncoupling electron transport and ATP synthesis, etc. This will help the reader understand the significance of the functional responses.
- Check that abbreviations are defined. For example, why is DMEM and HBSS not defined while FBS is?
- I see how animals were irradiated (section 4.3). I cannot find an analogous description for cells. It is not in section 4.1, despite the subsection title.
- Section 4.3 - “7.5 Gy of radiation (3 Gy/min) for 5 days.” Please clarify, is 7.5 Gy the total dose after 5 days, or is it 7.5 Gy per day for 5 days (37.5 Gy total!)? Thank for providing dose rates (Gy/min).
- What is the energy or wavelength of the radiation? Is the radiation intensity at the sample known?
- Is thermal damage a problem, esp. in cultures?
- Does the effect of radiation in vitro depend on the morphology of the cell culture? Confluence, layering, volume of media, etc. Does CON control for this?
- The type of damage observed is not surprising. You conclude that adipocytes and adipose tissue are “highly susceptible” to radiation damage. How does this susceptibility compare with other cell or tissue types? Please include a discussion comparing the damage observed, and if possible the susceptibility of damage, to other cell or tissue types.
Author Response
1. Because there are many materials and methods are used in the manuscript, it would be helpful to explicitly state at the start of each Results subsection or in each figure caption where the methods are found (i.e., which subsection(s) of the Materials and Methods section).
Many of the figures use various materials and methods, which would significantly lengthen the results or figure captions, so we have chosen not to do this.
2. In section 2.1, how is viability being quantified?
Trypan blue exclusion was used to assess viability. This has been made clear on page 2, line 88 of the manuscript.
3. Figure 1 – what does “2° only” mean?
For clarity, all images have been altered to say “no primary” as these have no primary antibody to demonstrate the specificity of staining.
4. Figure 1 - “A minimum of six fields/biological replicates (n=3) were averaged for quantification.” Does this mean six (6) fields of view each for three (3) independently prepared samples? Please rephrase for clarity. Is this true for all figures? How were fields of view chosen?
We have revised the statement for clarity: “Six random fields of view per independent replicate (n=3) were averaged for quantification.” This has been rephrased for clarity. Page 3 Line 100.
5. Why are DAPI images shown in the second column in some figures (e.g., Figure 1) and first column in other images (e.g., Figure 5)? If there is no reason, please be consistent. It will help the reader navigate the many figures.
All figures have been revised with DAPI in the second column.
6. The meaning of *** is missing in the caption for Figure 3. (I did not check all figures.) If the meaning of *, **, and *** is the same for all figures, say so.
** = p≤0.05 has been added to the caption of Figure 1 (page 3, lines 101-102). *** = p≤0.001 has been added to the caption for Figures 2 (page 5, lines 142-143) and 3 (Page 6, lines 173-174).
7. P11, line 248-250 – “In mice that received pelvic radiation, serum free glycerol [Figure 7B] and free fatty acids [Figure 7C] were elevated compared with the serum of unirradiated mice; however, only free glycerol levels were significantly elevated.” This sentence is confusing, the second part in particular. Why is glycerol mentioned twice? Are you concluding that levels are elevated or not?
The statement was revised to say: “In mice that received pelvic radiation, serum free glycerol was significantly elevated [Figure 8B] and free fatty acids [Figure 8C] trended to be enhanced as compared with the serum of unirradiated mice.” (page 11, line 268-270)
8. Figure 7 D, E – many functional modifiers are added to the sample. Please remind the reader of their effects, e.g., oligomycin decreases OCR by blocking ATP synthase proton channel, FCCP increases OCR by uncoupling electron transport and ATP synthesis, etc. This will help the reader understand the significance of the functional responses.
The effects of the modifiers were added to section 2.7 (page 11, lines 280-291).
9. Check that abbreviations are defined. For example, why is DMEM and HBSS not defined while FBS is?
Definitions for abbreviations were added for DMEM and HBSS. (Page 17, line 447; page 18, line 515)
10. I see how animals were irradiated (section 4.3). I cannot find an analogous description for cells. It is not in section 4.1, despite the subsection title.
“Starting on day 10, 3 Gy of radiation was administered daily for 3 days to the adipocytes using a RadSource 2000 X-Ray Box Irradiator at 1.2 Gy/min. Cells received a total dose of 9 Gy of radiation. “ Added for clarity on page 17, lines 451-453.
11. Section 4.3 - “7.5 Gy of radiation (3 Gy/min) for 5 days.” Please clarify, is 7.5 Gy the total dose after 5 days, or is it 7.5 Gy per day for 5 days (37.5 Gy total!)? Thanks for providing dose rates (Gy/min).
Total radiation dose was added to section 4.3, page 17, line 464.
12. What is the energy or wavelength of the radiation? Is the radiation intensity at the sample known?
The energy of the radiation is 320 kV. Dosimetry is completed every 6 months on the machine to guarantee that 1.2 Gy/min is being delivered to the samples during the radiation.
13. Is thermal damage a problem, esp. in cultures?
Thermal damage is not a problem; the irradiators used do not induce heat in the samples.
14. Does the effect of radiation in vitro depend on the morphology of the cell culture? Confluence, layering, volume of media, etc. Does CON control for this?
The 3T3 cells are at 100% confluency at the point of differentiation and then do not proliferate. We maintained a monolayer at 100% confluency throughout the experimental process. The volume of media was kept consistent at 2 mL of medium per well.
15. The type of damage observed is not surprising. You conclude that adipocytes and adipose tissue are “highly susceptible” to radiation damage. How does this susceptibility compare with other cell or tissue types? Please include a discussion comparing the damage observed, and if possible the susceptibility of damage, to other cell or tissue types.
The rational for adipose tissue being highly susceptible to radiation damage has been included in the discussion (page 16, lines 421-432).
Reviewer 2 Report
Comments and Suggestions for Authors
The manuscript of Liermann-Wooldrika et al, presents an interesting and novel investigation into the impact of radiation therapy on adipose tissue. The study aims to elucidate the oxidative stress, metabolic dysfunction, and inflammatory responses induced by radiation exposure in adipocytes, both in vitro and in vivo. The authors employ a combination of advanced techniques, including immunofluorescence, RNA sequencing, and metabolic assays, to demonstrate significant alterations in adipose tissue following irradiation. The research addresses a critical gap in the current understanding of how radiation therapy, a common treatment for various cancers, affects adipose tissue. The focus is on chronic damage, particularly with respect to metabolic dysfunction and immune infiltration and the novelty of this work lies in its exploration of clinically relevant radiation doses in vitro and in vivo models to demonstrate the persistence of damage in adipose tissue.
However, several methodological concerns need to be addressed, particularly regarding the image magnification, cell viability assessment, lipid content measurement, and immune cell characterization/distribution. These adjustments will enhance the manuscript’s impact and improve the overall clarity and reproducibility of the findings.
Major concerns
1)Figures: The images in Figures 4 and 5 have low magnification. The bright red staining seen in the controls should not be lost in the irradiated samples, suggesting that there may be a change in the microscope setup used for imaging. The images in Figure 4E show lymphocytic infiltration, Does it appear more prominent in the perivascular area??. To draw stronger conclusions, higher magnification is needed to assess the localization of oxidative DNA/RNA damage and lipid peroxidation at the subcellular level. Additionally, images with greater magnification will help clarify where these markers are accumulating within the adipocyte. Increasing magnification would allow more robust conclusions about the site of oxidative damage and peroxidation. Also, using cryosections instead of paraffin embedding may be beneficial, as the paraffin embedding process could disrupt lipid droplets and lead to tissue distortion, which could affect the results and interpretation.
2)Cell viability assessment: The manuscript mentions that the viability of adipocytes was not altered by radiation, but the method used to measure this is not clear. It is crucial to specify how viability was assessed in irradiated and control adipocytes (e.g., using viability assays such as MTT or trypan blue exclusion). This would ensure that the results are valid and provide clarity on whether the observed changes in lipid metabolism and immune infiltration are due to cell death or other sublethal damage.
3)Lipid content and metabolic measurements: The manuscript shows that lipid droplet size decreases in irradiated adipocytes, but the BODIPY and Oil Red O images do not provide conclusive evidence of this change. The bright red staining of the lipid droplets in the controls should not disappear in the irradiated samples. It is likely that there is a change in the setup used for imaging. To support these findings, it is recommended to measure the neutral lipid content (triglycerides, diacylglycerols) and free fatty acids in control and irradiated cells. This could offer a more comprehensive understanding of lipid metabolism alterations following radiation exposure.
4)In Figure 5, the immune cells (CD4+ T-cells and CD8+ T-cells) appear to localize around the lipid droplets. The circumferential staining pattern around the lipid droplets is worth investigating, as it could be an artifact of the tissue preparation technique (paraffin embedding). The tissue disruption caused by paraffin embedding could affect the infiltration pattern, and using cryosections might eliminate this potential artifact.
Author Response
1)Figures: The images in Figures 4 and 5 have low magnification. The bright red staining seen in the controls should not be lost in the irradiated samples, suggesting that there may be a change in the microscope setup used for imaging. The images in Figure 4E show lymphocytic infiltration, Does it appear more prominent in the perivascular area??. To draw stronger conclusions, higher magnification is needed to assess the localization of oxidative DNA/RNA damage and lipid peroxidation at the subcellular level. Additionally, images with greater magnification will help clarify where these markers are accumulating within the adipocyte. Increasing magnification would allow more robust conclusions about the site of oxidative damage and peroxidation. Also, using cryosections instead of paraffin embedding may be beneficial, as the paraffin embedding process could disrupt lipid droplets and lead to tissue distortion, which could affect the results and interpretation.
In Figures 4, 5, and 6, we enlarged the images to better visualize the fluorescence staining. In paraffin-embedded samples, the lipids are removed from the tissues, but the components surrounding the lipid droplet remain. Since the majority of an adipocyte is the lipid droplet, all other cellular components are found in the periphery, making subcellular localization difficult. Given the turnaround time for this manuscript, we are not able to collect cryosections and re-stain the tissue samples.
2)Cell viability assessment: The manuscript mentions that the viability of adipocytes was not altered by radiation, but the method used to measure this is not clear. It is crucial to specify how viability was assessed in irradiated and control adipocytes (e.g., using viability assays such as MTT or trypan blue exclusion). This would ensure that the results are valid and provide clarity on whether the observed changes in lipid metabolism and immune infiltration are due to cell death or other sublethal damage.
Trypan blue exclusion was used to assess viability. This has been made clear on page 2, line 88 of the manuscript. Cell viability was added to the methods section (section 4.6, page 18).
3)Lipid content and metabolic measurements: The manuscript shows that lipid droplet size decreases in irradiated adipocytes, but the BODIPY and Oil Red O images do not provide conclusive evidence of this change. The bright red staining of the lipid droplets in the controls should not disappear in the irradiated samples. It is likely that there is a change in the setup used for imaging. To support these findings, it is recommended to measure the neutral lipid content (triglycerides, diacylglycerols) and free fatty acids in control and irradiated cells. This could offer a more comprehensive understanding of lipid metabolism alterations following radiation exposure.
The loss of bright red staining in the irradiated lipid droplets is attributed to the increase in lipolytic activity. The fluorescence signal decreases as the adipocytes lose lipids during lipolysis. To corroborate this data, media was collected from irradiated and unirradiated adipocytes to show that the BODIPY-stained lipids are secreted into the media following radiation. This experiment was added to new Figure 2B (section 2.2, page 4, lines 108-114) and the methodology (section 4.8, page 19, lines 552-558).
4)In Figure 5, the immune cells (CD4+ T-cells and CD8+ T-cells) appear to localize around the lipid droplets. The circumferential staining pattern around the lipid droplets is worth investigating, as it could be an artifact of the tissue preparation technique (paraffin embedding). The tissue disruption caused by paraffin embedding could affect the infiltration pattern, and using cryosections might eliminate this potential artifact.
Figure 5 has been further enhanced to better showcase immune infiltration. Crown-like structures, or macrophages surrounding the outer surface of adipocytes, are a well-characterized feature of inflamed adipose tissue. Therefore, the staining pattern of the immune cells around the lipid droplet, or the circumference of the adipocyte, is not surprising, but rather similar to other published reports [1].
- Choi, C., et al., TM4SF19-mediated control of lysosomal activity in macrophages contributes to obesity-induced inflammation and metabolic dysfunction. Nat Commun, 2024. 15(1): p. 2779.
Round 2
Reviewer 2 Report
Comments and Suggestions for Authors
The authors have done an excellent job addressing most of the concerns raised. The manuscript has been improved substantially in clarity.
However, one important methodological issue remains in Sections 4.1 ("Cell Culture and In Vitro Radiation") and 4.8 ("Lipolytic Activity Methodology"). Specifically, the term “conditioned medium” mentioned in Section 4.8 is not clearly defined in Section 4.1. It is inferred that this medium contains 10% fetal bovine serum (FBS), but this should be explicitly stated.
This detail is critical because the use of FBS-containing medium could significantly influence the interpretation of the BODIPY-based lipolysis assay. The presence of lipid acceptors in the serum likely facilitates the diffusion of BODIPY-labeled fatty acids from the adipocytes into the medium. As a result, the observed decrease in cellular fluorescence may not represent active lipolysis but rather passive efflux due to the presence of serum components. Therefore, this experimental design does not convincingly demonstrate fatty acid release. To provide stronger evidence of lipolytic activity, the authors should complement this assay by directly quantifying free fatty acids and glycerol in the medium and/or measuring intracellular triglyceride levels. These standard biochemical measurements would more reliably support the conclusion that radiation induces lipolysis in adipocytes.
Author Response
All changes have been made in blue for the second revision.
However, one important methodological issue remains in Sections 4.1 ("Cell Culture and In Vitro Radiation") and 4.8 ("Lipolytic Activity Methodology"). Specifically, the term “conditioned medium” mentioned in Section 4.8 is not clearly defined in Section 4.1. It is inferred that this medium contains 10% fetal bovine serum (FBS), but this should be explicitly stated.
“DMEM containing 10% FBS and 1% penicillin/streptomycin” was added to Section 4.1, page 17, line 456.
This detail is critical because the use of FBS-containing medium could significantly influence the interpretation of the BODIPY-based lipolysis assay. The presence of lipid acceptors in the serum likely facilitates the diffusion of BODIPY-labeled fatty acids from the adipocytes into the medium. As a result, the observed decrease in cellular fluorescence may not represent active lipolysis but rather passive efflux due to the presence of serum components. Therefore, this experimental design does not convincingly demonstrate fatty acid release. To provide stronger evidence of lipolytic activity, the authors should complement this assay by directly quantifying free fatty acids and glycerol in the medium and/or measuring intracellular triglyceride levels. These standard biochemical measurements would more reliably support the conclusion that radiation induces lipolysis in adipocytes.
We conducted a free fatty acid measurement in the media from adipocytes exposed to radiation or sham radiation. We found that free fatty acids were significantly elevated in the media. This data is now found in new Figure 2D.
Round 3
Reviewer 2 Report
Comments and Suggestions for Authors
All concerns were addressed, good piece of work.